# Genome-Wide Identification of the *HMA* Gene Family and Expression Analysis under Cd Stress in Barley

**DOI:** 10.3390/plants10091849

**Published:** 2021-09-06

**Authors:** Chiran Zhang, Qianhui Yang, Xiaoqin Zhang, Xian Zhang, Tongyuan Yu, Yuhuan Wu, Yunxia Fang, Dawei Xue

**Affiliations:** College of Life and Environmental Sciences, Hangzhou Normal University, Hangzhou 311121, China; zhangchiran2021@126.com (C.Z.); qhyangqianhui@126.com (Q.Y.); zxq@hznu.edu.cn (X.Z.); zhangxian@hznu.edu.cn (X.Z.); yuty1223@163.com (T.Y.); yuhuanwu@hznu.edu.cn (Y.W.)

**Keywords:** barley, *HMA* gene family, bioinformatics analysis, Cd stress

## Abstract

In recent years, cadmium (Cd) pollution in soil has increased with increasing industrial activities, which has restricted crop growth and agricultural development. The heavy metal ATPase (*HMA*) gene family contributes to heavy metal stress resistance in plants. In this study, 21 *HMA* genes (*HvHMAs*) were identified in barley (*Hordeum*
*vulgare* L., *Hv*) using bioinformatics methods. Based on phylogenetic analysis and domain distribution, barley *HMA* genes were divided into five groups (A–E), and complete analyses were performed in terms of physicochemical properties, structural characteristics, conserved domains, and chromosome localization. The expression pattern analysis showed that most *HvHMA* genes were expressed in barley and exhibited tissue specificity. According to the fragments per kilobase of exon per million fragments values in shoots from seedlings at the 10 cm shoot stage (LEA) and phylogenetic analysis, five *HvHMA* genes were selected for expression analysis under Cd stress. Among the five *HvHMA* genes, three (*HvHMA1*, *HvHMA3*, and *HvHMA4*) were upregulated and two (*HvHMA2* and *HvHMA6*) were downregulated following Cd treatments. This study serves as a foundation for clarifying the functions of HvHMA proteins in the heavy metal stress resistance of barley.

## 1. Introduction

Cadmium (Cd) pollution is one of the negative consequences of industrialization. Cd is highly toxic to plants and is easily absorbed by the roots and accumulates in the tissues [1], which influences various processes including water and mineral uptake, respiration, and photosynthesis, and leads to the inhibition of growth and even death [2]. Cd ions with a lack of specificity enter the plant through other transporters (Fe^2+^/Fe^3+^, Zn^2+^, and Mn^2+^) and compete with other nutrients for plant uptake, resulting in deficient nutrition [3,4]. In response to Cd poisoning, various defense mechanisms have evolved in plants, such as extrusion across plasma membrane, chelation in the cytosol, and vacuolar sequestration [5]. Previous studies have identified multiple proteins related to Cd transport, including heavy metal ATPase (HMA) [6], yellow stripe-like proteins (YSL) [7], and natural resistance-associated macrophage proteins (NRAMP) [8], to name a few.

HMA, also known as P_1B_-ATPase, is a type of protein combining ATP hydrolysis with metal ion transport across membranes [9,10] participating in absorbing and transporting heavy metal ions (Cu^2+^, Zn^2+^, Co^2+^, Cd^2+^, and Pb^2+^) [6]. Typical HMA proteins contain the E1–E2 ATPase domain and haloacid dehalogenase-like hydrolase (Hydrolase) domain [11]. Additionally, both sides of the N-terminal and C-terminal metal-binding sites may possess one or more soluble metal-binding domains (MBDs) that interact with or bind to specific metal ions [6]. At present, a number of *HMA* genes have been identified in plants, including 8 in *Arabidopsis thaliana* [12], 9 in rice (*Oryza sativa* L.) [12] 11 in maize (*Zea mays* L.) [13], 11 in sorghum (*Sorghum bicolor* L.) [13], 17 in *Populus trichocarpa* [11], and 20 in soybean (*Glycine max* L.) [14]. Studies have demonstrated that *AtHMA2* and *AtHMA4* are two essential genes mediating Cd translocation in *A. thaliana* [15]. The translocation of Cd from the roots to shoots was near-completely abolished in the *hma2 hma4* double mutant. TcHMA3, a tonoplast-localized transporter highly specific for Cd, is responsible for sequestering Cd into the leaf vacuoles so as to detoxify Cd in *Thlaspi caerulescens* [16]. OsHMA3, which localizes to vacuolar membranes, was identified as the gene that controls root-to-shoot Cd translocation rates in rice [17]. These results indicate that the *HMA* gene family plays diverse roles in plant resistance to Cd stress.

Barley (*Hordeum vulgare* L., *Hv*), an important cereal crop, is widely used in numerous industries, including animal feed, brewing, and food [18]. The exploration of vital genes related to heavy metal stress resistance is beneficial for cultivating Cd-tolerant barley varieties. Recent barley genome sequencing accomplishments have facilitated further studies on barley genomics. Studies on the *SBP-box* [19], *WRKY* [20], *ABC* [21], *F-box* [22], and *SOD* [23] gene family in barley have been successfully completed. Although the functions of several HMA proteins in barley have been reported [24,25], genome-wide analysis of the *HvHMA* family is lacking. In this study, the *HvHMA* gene family was genome-widely identified in barley, and the phylogenetic relationships, structural characteristics, physicochemical properties, chromosomal location, as well as the tissue expression of identified members were analyzed. Moreover, the expression of some members following Cd treatment was investigated using quantitative real-time polymerase chain reaction (qRT-PCR). These combined analyses of the biological characteristics and expression changes of the *HvHMA* gene family provide helpful information for studying the function of *HvHMA* genes and improve the Cd tolerance of barley varieties.

## 2. Materials and Methods

### 2.1. Plant Materials and Treatment

The ‘ZJU3’ barley variety was used in this study. Seeds uniform in size and with a full shape were selected and sterilized in 2.5% NaClO for 10 min, rinsed with distilled water four times, and then germinated at 28 °C under dark conditions. After 48 h, seedlings with a root length of approximately 0.5 cm were moved to hydroponic culture boxes (day/night temperatures of 26 °C/24 °C, light/dark photoperiod of 14 h/10 h, and light intensity of 18000 Lx). At the one-leaf stage, the seedlings were treated with 1/4 Hoagland’s nutrient solution. At the two-leaf stage, Cd stress experiments were performed. The CdCl_2_ solutions (50 μmol/L and 100 μmol/L) prepared with Hoagland’s nutrient solution were used to simulate Cd stress, and Hoagland’s nutrient solution without CdCl_2_ was used as the control. After 120 h of treatment, more than 10 barley seedlings were selected for each sample, and quickly stored at −80 °C until analysis. The experiment was performed in triplicate. The Hoagland’s nutrient solution formula was as described by Zhang et al. [26].

### 2.2. RNA Isolation and cDNA Synthesis

The total RNA was isolated from barley leaves using an RNA extraction kit (Tiangen, Beijing, China) and reverse transcribed to generate cDNA using a reverse transcription kit (Yeasen, Shanghai, China). The cDNA obtained was stored at −20 °C for qRT-PCR analysis.

### 2.3. Bioinformatics Analysis of the Barley HMA Gene Family

#### 2.3.1. Identification and Structural Analysis of Barley HMA Genes

The HMA protein sequences of *A. thaliana* and rice were obtained from TAIR (https://www.arabidopsis.org/index.jsp; accessed on 3 July 2020) and RiceData (https://www.ricedata.cn/gene/; accessed on 3 July 2020), respectively. The HMMER profiles related to the conserved domains of HMA proteins (E1–E2 ATPase: PF00122; Hydrolase: PF00702) were downloaded from the Pfam database (http://pfam.xfam.org/; accessed on 9 July 2020) [27]. First, the candidate protein sequences were uploaded onto the CD-HIT website (http://weizhong-lab.ucsd.edu/cdhit-web-server/cgi-bin/index.cgi?cmd=cd-hit; accessed on 16 July 2020) [28] and SMART Web server (http://smart.embl-heidelberg.de/; accessed on 17 July 2020) [29] to remove sequences without E1-E2 ATPase and Hydrolase domains. Afterward, the non-redundant barley HMA proteins were obtained by manually removing the redundant sequences. The batch sequence search function in the Pfam database was used to obtain gene annotation files, and TBtools v1.087 (Chen, C.C., South China Agricultural University (SCAU), Guangdong, China) was then used to draw the domain map. The molecular characteristics of the HvHMA proteins were analyzed in ExPASy (Compute pI/Mwtool) (https://web.expasy.org/protparam/; accessed on 26 July 2020) including the number of amino acid (aa) residues, molecular weight (MW), theoretical isoelectric point (pI), and grand average of hydropathicity (GRAVY). WoLF PSORT (https://wolfpsort.hgc.jp/; accessed on 31 July 2020) [30] was used to predict the subcellular localization of the HvHMA proteins. The conserved motifs of the HvHMA proteins were mapped using the MEME online tool (http://meme-suite.org/tools/meme; accessed on 4 August 2020) [31]. The intron-exon organizations of the *HvHMA* genes were generated using Gene Structure Display Server v2.0 (GSDS v2.0, http://gsds.cbi.pku.edu.cn/; accessed on 6 August 2020) (Center for Bioinformatics (CBI), Beijing, China) [32] by comparing the cDNAs to their corresponding genomic DNA sequences. The *HvHMA* genes were mapped to barley chromosomes based on physical location information from the EnsemblPlants database (http://plants.ensembl.org/Hordeum_vulgare/Info/Index; accessed on 6 August 2020) using Tbtools v1.087.

#### 2.3.2. Phylogenetic Analysis of the Barley HMA Family

The HMA protein sequences of *A. thaliana*, rice, and barley were imported into the MEGA v7.0 program (Sudhir Kumar, Temple University, Philadelphia, PA, USA) and multiple sequence alignments were performed using ClustalW. The alignment file was then used to construct a neighbor-joining (NJ) phylogenetic tree, with the following parameters: p-distance model, 1000 bootstrap replications, and other default parameters [33]. The tree was displayed and modified using the iTOL website (https://itol.embl.de/; accessed on 12 August 2020) [34].

#### 2.3.3. Expression Profiling of the Barley HMA Family and Candidate Gene Selection

The RNA-Seq data of 15 developmental stages were downloaded from the IPK website (https://apex.ipk-gatersleben.de/apex/f?p=284:10; accessed on 20 August 2020) for mapping *HvHMA* expression profiles. The 15 developmental stages were as follows: 4-day embryos (EMB); roots from seedlings (10 cm shoot stage) (ROO1); shoots from seedlings (10 cm shoot stage) (LEA); developing inflorescences (INF2); developing tillers, 3rd internode (5 DAP) (NOD); developing grain (5 DAP) (CAR5); developing grain (15 DAP) (CAR15); etiolated seeding, dark conditions (10 DAP) (ETI); inflorescences, lemma (42 DAP) (LEM); inflorescences, lodicule (42 DAP) (LOD); dissected inflorescences (42 DAP) (PAL); epidermal strips (28 DAP) (EPI); inflorescences, rachis (35 DAP) (RAC); roots (28 DAP) (ROO2); and senescing leaves (56 DAP) (SEN). The expression of *HvHMA* genes was normalized and represented in fragments per kilobase of exon per million fragments mapped (FPKM). The *HvHMA* expression profile based on the FPKM values was then drawn using the Multiple Experiment Viewer (MeV) (J. Craig Venter Institute, La Jolla, CA, USA) [35]. Based on the FPKM values at the 10 cm shoot stage and phylogenetic analysis, the candidate genes were selected for qRT-PCR experiments.

### 2.4. Quantitative RT-PCR Analysis of Barley HMA Genes

Five pairs of primers related to specific genes were designed using Primer Premier v5.0 (PREMIER Biosoft, San Francisco, CA, USA) for qRT-PCR (Table 1). The barley actin gene *HvActin* (*HORVU1Hr1G002840*) was used as an internal control. The qRT-PCR analysis was performed on the CFX96 Real-Time PCR Detection System (Bio-Rad, Hercules, CA, USA), and the data were analyzed using the 2^−ΔΔCt^ method with three biological replicates [36]. IBM SPSS Statistics v20 (IBM, Armonk, NY, USA) statistical software was then used to analyze significance (*, **, and *** indicates *p* < 0.05, *p* < 0.01, and *p* < 0.001 respectively). Histograms were drawn with SigmaPlot v10.0 (SYSTAT, San Jose, CA, USA).

## 3. Results

### 3.1. Identification and Molecular Characteristics of Barley HMA Proteins

Through multiple bioinformatics analyses, a total of 21 barley HMA proteins were screened by removing redundant sequences and validating domains, which were named HvHMA1-21. The basic information on HvHMA1-21, including the number of amino acid residues, molecular weight, theoretical isoelectric point, grand average of hydropathicity, and subcellular localization, which indicated molecular characteristics of barley HMA proteins, is listed in Table 2. The results showed that the 21 HvHMA proteins contained 672 (HvHMA5) to 1083 (HvHMA21) amino acid residues with molecular weights ranging from 73112.29 (HvHMA5) to 118116.93 (HvHMA21) Da. The theoretical isoelectric points of the 21 HvHMA proteins ranged from 5.00 (HvHMA8) to 7.82 (HvHMA17), with the majority constituting acidic proteins. The GRAVY numeric values of the 21 HvHMA proteins varied from −0.090 (HvHMA1) to 0.422 (HvHMA3), indicating that these proteins were likely amphoteric proteins. Additionally, the subcellular localization results showed that the 18 HvHMA proteins were localized in the plasma membrane, whereas three proteins (HvHMA9, HvHMA14, and HvHMA19) were localized in the endoplasmic reticulum.

### 3.2. Phylogenetic Analysis and Classification of Barley HMA Genes

To explore the evolutionary characteristics of *HvHMA* genes and the evolutionary relationships between the *AtHMA*, *OsHMA*, and *HvHMA* genes, HMA sequences from *A. thaliana*, rice, and barley, including 8 AtHMA proteins, 8 OsHMA proteins, and 21 HvHMA proteins, were subjected to phylogenetic analysis. As shown in Figure 1, the *HvHMA* genes were divided into five groups (A–E). Among the 21 *HvHMA* genes, 2 belong to group A, 6 to group B, 6 to group C, 3 to group D, and 4 to group E. The phylogenetic tree indicated that members of groups A, D, and E were homologous to the AtHMA and OsHMA proteins. Moreover, compared to dicotyledonous *A. thaliana*, monocotyledonous barley and rice were more closely related. With the exception of the E1-E2 ATPase and Hydrolase domains, some HvHMA proteins contained other domains including an HMA domain, Cation_ATPase_N domain, Cation_ATPase_C domain, Cation_ATPase domain, Hydrolase_3 domain, and CaATP_NAI domain, which revealed that the HvHMA proteins contained more abundant domains than the AtHMA and OsHMA proteins. Thus, it is inferred that members of the barley *HMA* gene family are more functionally diverse and therefore worth exploring. Additionally, there were some differences among groups in the domain distribution of the HvHMA proteins. The HMA domains were concentrated in groups D and E, the Cation_ATPase_N domains and Cation_ATPase domains were distributed in groups B and C, and Cation_ATPase_C domains were all distributed in group B. In relation to other groups, there were more types of domains in group B, indicating that members of group B might be complex in terms of function.

### 3.3. Chromosomal Location of Barley HMA Genes

According to the genome annotations, 18 of the 21 *HvHMA* genes were distributed on the six barley chromosomes (Figure 2), with the largest number of genes located on Chr4 (5), followed by Chr5 (4), Chr7 (4), Chr1 (2), Chr2 (2), and Chr6 (1). However, *HvHMA7*, *HvHMA12*, and *HvHMA16*, without clear localization information, were not positioned onto barley chromosomes. In addition, most of the *HvHMA* genes were concentrated on or near the end of the barley chromosomes. These results suggested that the distribution of *HvHMA* genes was uneven.

### 3.4. Motif Composition of the Barley HMA Proteins

Conserved motif analysis of the HvHMA proteins helped elucidate the conservation as well as the diversification of this family, and a total of 10 distinct conserved motifs were revealed. As exhibited in Figure 3, all HvHMA proteins contained common motifs including motif 1 and motif 10, which suggested that the two motifs might be the characteristic motifs of HvHMA proteins. With the exception of HvHMA21, all HvHMA proteins contained motif 3. Additionally, HvHMA proteins within the same group were generally found to show a similar motif composition. For example, motif 2 was distributed in all groups except group A, whereas motif 4, motif 5, motif 6, motif 8, and motif 9 were unique to group C. Moreover, the differences in motif composition among groups combined with the phylogenetic analysis results supported the reliability of the group classifications and indicated that *HvHMA* genes in distinct groups might be functionally divergent.

### 3.5. Intron-Exon Structure of Barley HMA Genes

The introns disrupted the coding sequences of most *HvHMA* genes. As shown in Figure 4, there were some differences among the *HvHMA* genes in terms of the number and size of the introns, which might be caused by intron deletion and insertion events. With the exception of *HvHMA16* without introns, all *HvHMA* genes contained 2–33 introns (13 with 2–8 introns, 6 with 12–20 introns, and 1 with 33 introns). Additionally, several non-coding regions were distributed in the 18 *HvHMA* genes, with the exception of *HvHMA16, HvHMA18,* and *HvHMA20.*

### 3.6. Expression Pattern Analysis of Barley HMA Genes and Target Gene Screening

The analysis of gene expression patterns contributed to gene function prediction. The expression profiles of the *HvHMA* genes (Figure 5) revealed that the expression of some genes was tissue-specific. For example, the expression levels of *HvHMA5*, *HvHMA9*, and *HvHMA15* were high during grain development; the expression levels of *HvHMA3*, *HvHMA6*, *HvHMA7*, *HvHMA8*, *HvHMA13,* and *HvHMA18* were high in the leaves; *HvHMA2*, *HvHMA12**,* and *HvHMA19* were specifically expressed in the inflorescences; and *HvHMA4*, *HvHMA10*, and *HvHMA11* were specifically expressed in the tillers, roots, and epidermal strips, respectively. These results indicated that these genes might play specific roles in the relevant tissues. Moreover, the clustering results of the expression data unclearly correspond to the groupings based on phylogenetic analysis, implying that the expression pattern similarity incompletely depended on the sequence similarity. Based on a comprehensive consideration of the FPKM values at the 10 cm shoot stage and phylogenetic analysis, a total of five *HvHMA* genes were screened for expression analysis under Cd stress.

### 3.7. Expression Analysis of Barley HMA Genes in Response to Cd Treatment

HMA proteins participate in the distribution of non-essential heavy metal ions including Cd^2+^, which greatly affect the plant response to heavy metal stress. To analyze the expression of *HvHMA* genes under Cd stress, five members (Table 1) were carefully selected from 21 *HvHMA* genes, and qRT-PCR experiments were further performed at the seedling stage. The results (Figure 6) revealed that three genes (*HvHMA1*, *HvHMA3*, and *HvHMA4*) were upregulated and two genes (*HvHMA2* and *HvHMA6*) were downregulated. Compared with the control, the expression levels of all five genes were significantly different under the 100 μmol/L CdCl_2_ treatment, whereas four genes, except for *HvHMA4,* were significantly different under the 50 μmol/L CdCl_2_ treatment. With the increase of Cd concentration, the expression of *HvHMA1* and *HvHMA4* was significantly higher under the high-concentration stress than under the low-concentration stress. Moreover, the expression changes of *HvHMA2*, *HvHMA3*, and *HvHMA6* were similar after the two Cd treatments: the expression of *HvHMA2*, *HvHMA3*, and *HvHMA6* was slightly higher under the high-concentration stress than under the low-concentration stress.

## 4. Discussion

The *HMA* family, which plays a significant role in heavy metal transport, exists widely in plants. In this study, 21 *HMA* genes were identified in barley. According to phylogenetic analysis, the 21 *HvHMA* genes could be divided into five groups (A–E). Compared to groups B and C, members belonging to groups A, D, and E possessed higher homology to the proteins of *A. thaliana* and rice. Except for the E1-E2 ATPase and Hydrolase domains, members of groups B and C contained the Cation_ATPase_N domains. In addition, members of group B contained the Cation_ATPase_C domains. The Cation_ATPase_N domains and the Cation_ATPase_C domains, which are metal-binding domains, participate in metal ion transport in plants. As indicated by the conserved motif analysis, motifs 4–6 and motifs 8–9 were only distributed in group C. Therefore, it is speculated that the domains as well as motifs unique to groups B and C resulted in the separation of groups B and C from the other groups in the phylogenetic tree. Furthermore, the characteristics of groups B and C illustrated the differences in evolution between barley and other species.

The subcellular localization results revealed that most of the HvHMA members were predicted as plasma membrane proteins, with the exception of HvHMA9, HvHMA14, and HvHMA19, which were all located in the endoplasmic reticulum and were placed in the same group (group B). These results suggested that there were certain corresponding relationships between the phylogenetic groupings based on sequence similarity and subcellular localization. Therefore, homologous genes may be similar in gene function and signal transduction process.

According to the presence or absence of introns, eukaryotic genes can be divided into intron-containing and intronless genes. Most eukaryotic genes belong to the former, but some belong to the latter. Previous studies identified that there are 5846 (21.7%) intronless genes in *A. thaliana* and 11,109 (19.9%) in rice [37]. Among the 21 *HvHMA* genes, 20 members with 2–33 introns are intron-containing genes, whereas *HvHMA16* is an intronless gene. Some plausible explanations may account for the origin of intronless genes. It has been suggested that intronless genes evolved owing to a loss of introns [38]. Another probability is that intronless genes formed as a result of reverse transcription [39]. During the process of retroposition, mRNAs are reverse-transcribed into cDNAs and inserted into new genomic positions that lack introns [40]. Therefore, it can be inferred that intron loss or retrotransposition events impacted on the intron-exon structures of *HvHMA* genes, leading to the presence of an intronless gene (*HvHMA16*) in the barley *HMA* gene family. By comprehensively analyzing the results of the evolutionary tree and the expression values at the 10 cm shoot stage, five genes were screened that might be related to stress responses. Among them, the expressions of *HvHMA1*, *HvHMA3,* and *HvHMA4* were significantly upregulated under Cd stress. *HvHMA1* was highly homologous to *OsHMA2*, which participates in the root-to-shoot translocation of Cd [41,42,43]. Compared to the wild-type (WT), the Cd concentration in the grains of *OsHMA2*-overexpressing rice was decreased by approximately half [44]. *HvHMA3* was homologous to *OsHMA3*, which sequesters Cd into the vacuoles of root cells in rice, thereby controlling the rate of Cd translocation from the roots to shoots [45]. Additionally, *HvHMA4* was homologous to *OsHMA9,* whose expression was induced by a high concentration of Cd [46]. The above results indicated that the changes in expression after Cd treatment were in line with theoretical expectations and that the phylogenetic analysis results were credible.

Cd stress negatively effects plant growth and development. The qRT-PCR results suggested that Cd stress can promote or inhibit the expression of *HvHMA* genes, which indicated that different *HvHMAs* exhibit diverse mechanisms to protect barley from stress damage. The expression of *HvHMA1* gradually increased with the growth of Cd concentration. Mills [47] found that the corresponding *HvHMA* gene conferred Cd sensitivity to wild-type yeast due to transport activity. It was speculated that *HvHMA1* changed the transport activity of Cd in response to Cd stress in barley. Lei [48] found that *HvHMA3* played a crucial role in grain Cd accumulation. In this study, *HvHMA3* were significantly upregulated after Cd treatment. Therefore, it was speculated that *HvHMA3* might be involved in Cd distribution in stress condition. Following Cd stress, the expression levels of *HvHMA2* and *HvHMA6* decreased significantly, indicating that Cd stress negatively regulated the expression of the two genes. According to this, it is inferred that the expression regulation pathways related to *HvHMA2* and *HvHMA6* may be similar. Among experimental groups, *HvHMA1* and *HvHMA4* were significantly upregulated with increasing Cd concentration, which indicated that the two genes were sensitive to the change of Cd stress. However, the molecular mechanisms of these *HvHMA* genes in response to Cd stress need further exploration.

## 5. Conclusions

The barley *HMA* gene family was explored herein using bioinformatics analysis. The results revealed the characteristics of the barley *HMA* gene family in terms of physicochemical properties, phylogenetic relationships, domain distribution, chromosomal location, motif composition, intron-exon structure, as well as tissue expression. Moreover, five *HvHMA* genes were selected for expression analysis which indicated that the five genes responded differently to Cd stress. *HvHMA1*, *HvHMA3*, and *HvHMA4* were strongly activated by Cd stress, whereas *HvHMA2* and *HvHMA6* were significantly restrained. This study preliminarily confirms that *HvHMA1*, *HvHMA3*, and *HvHMA4* play vital roles in Cd tolerance, providing a theoretical basis for further research on the functions of related genes and the improvement of barley varieties.

## Figures and Tables

**Figure 1 plants-10-01849-f001:**
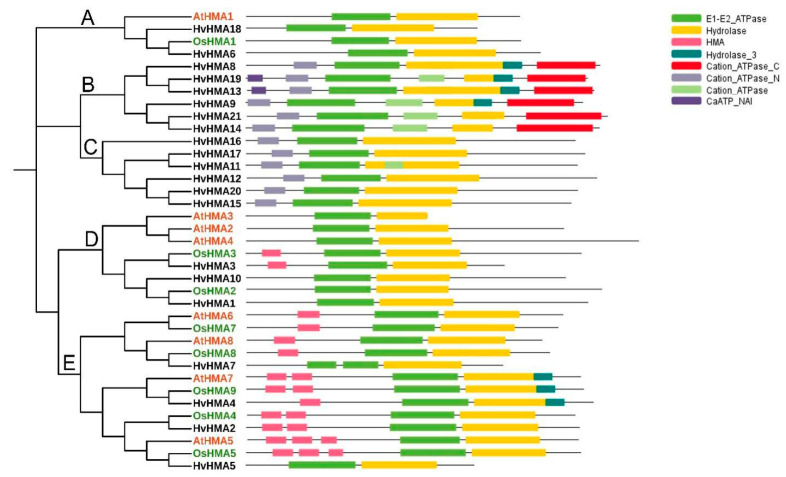
The phylogenetic tree was constructed using the HMA protein sequences of *Hordeum vulgare* (Hv), *Oryza sativa* (Os), and *Arabidopsis thaliana* (At) in MEGA v7.0. All HMA members were classified into five groups (A–E). The HMA proteins of different species are differentiated by different colors. The black letters represent HvHMAs, the green letters represent OsHMAs, and the red letters represent AtHMAs. Different-colored rectangles represent different structural domains. The three green rectangles from light to dark represent Cation_ATPase, E1–E2 ATPase, and Hydrolase_3, respectively; the yellow rectangle represents Hydrolase; the pink rectangle represents HMA; the red rectangle represents Cation_ATPase_C; and the two purple rectangles from light to dark represent Cation_ATPase_N and CaATP_NAI, respectively.

**Figure 2 plants-10-01849-f002:**
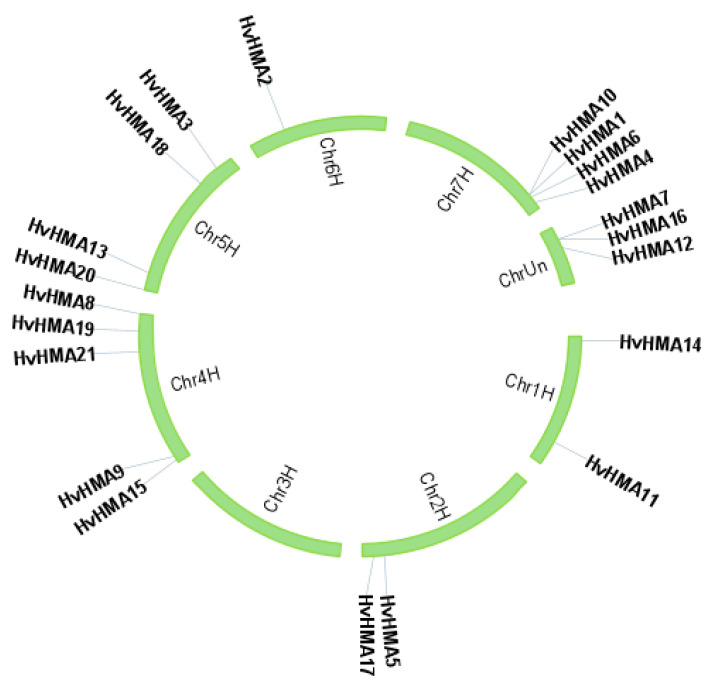
Distribution of *HvHMA* genes on barley chromosomes. The location information of the *HvHMA* genes was obtained from the EnsemblPlants database.

**Figure 3 plants-10-01849-f003:**
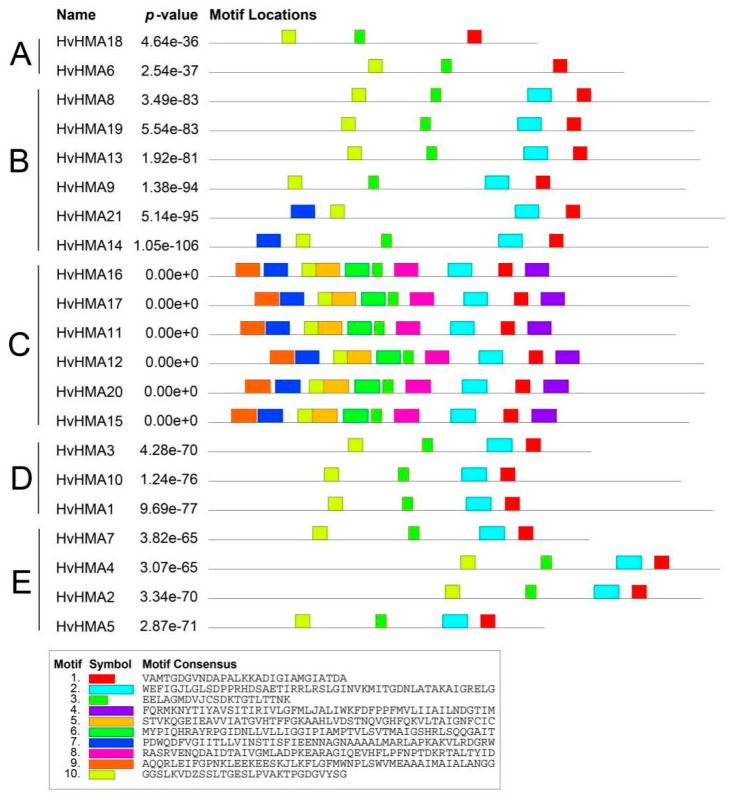
Motif distribution of the HvHMA family in barley. Different conservative motifs are represented by boxes of 10 different colors (color figure online).

**Figure 4 plants-10-01849-f004:**
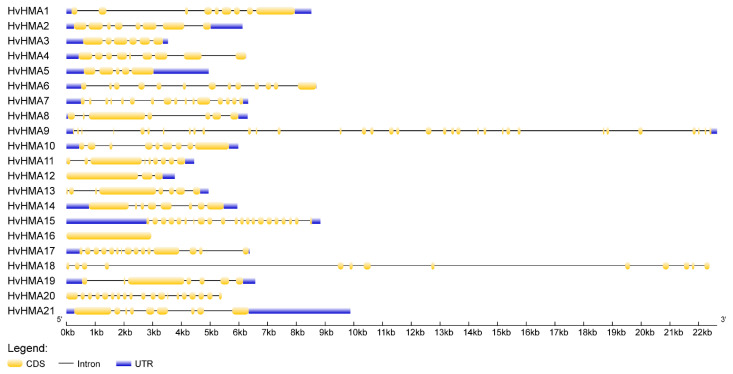
Intron-exon structures of *HvHMA* genes in barley. The exons and introns are indicated by yellow boxes and black lines, respectively (color figure online).

**Figure 5 plants-10-01849-f005:**
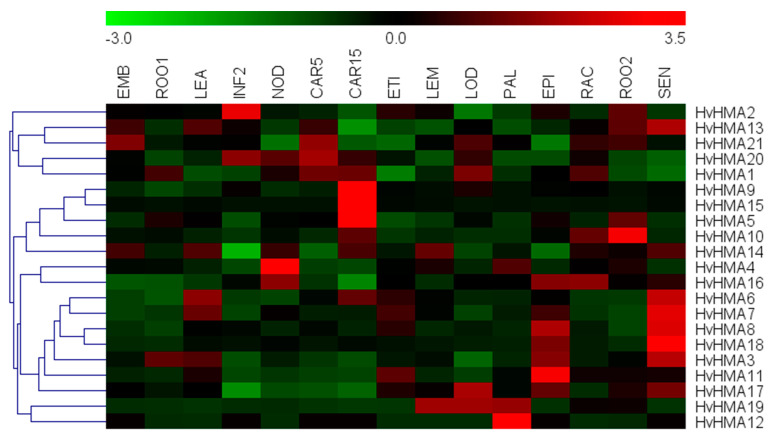
Expression profiles of *HvHMA* genes in different tissues and development stages. Data were obtained from a publicly available database. Columns represent *HvHMA* members, while rows show different developmental stages and tissues. The expression level of *HvHMAs* is shown by the intensity of the color, where red represents high expression and green represents low expression. EMB, 4-day embryos; ROO1, roots from seedlings (10 cm shoot stage); LEA, shoots from seedlings (10 cm shoot stage); INF2, developing inflorescences; NOD, developing tillers, 3rd internode (5 DAP); CAR5, developing grain (5 DAP); CAR15, developing grain (15 DAP); ETI, etiolated seeding, dark conditions (10 DAP); LEM, inflorescences, lemma (42 DAP); LOD, inflorescences, lodicule (42 DAP); PAL, dissected inflorescences (42 DAP); EPI, epidermal strips (28 DAP); RAC, inflorescences, rachis (35 DAP); ROO2, roots (28 DAP); and SEN, senescing leaves (56 DAP) (color figure online).

**Figure 6 plants-10-01849-f006:**
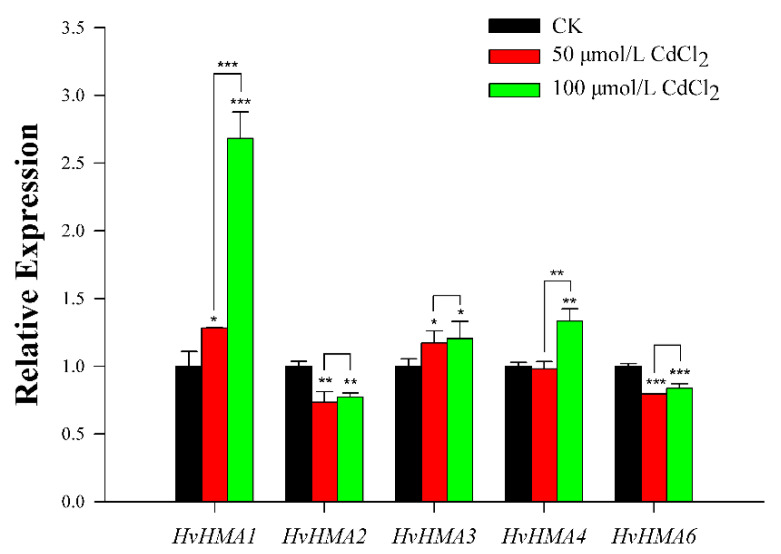
Relative expression analysis of five *HvHMA* genes under Cd stress in barley seedling leaves. Different treatments are represented by three different colors, and columns in black represent CK, columns in red represent 50 μmol/L CdCl_2_, and columns in green represent 100 μmol/L CdCl_2_. ANOVA was used to test significance. * indicates *p* < 0.05, ** indicates *p* < 0.01, and *** indicates *p* < 0.001. Error bars represent the standard deviation.

**Table 1 plants-10-01849-t001:** Primer sequences designed for qRT-PCR analysis.

Gene Name	Gene ID	Forward Primer Sequence (5′-3′)	Reverse Primer Sequence (5′-3′)
*HvHMA1*	*HORVU7Hr1G097240.1*	TGGCGAAGAAATGCTGTGCT	AACCGCCTGTTGATACATTCTC
*HvHMA2*	*HORVU6Hr1G033380.2*	TGGAGGTGTCATTTCAGAAGTGG	CAACACCATCAACTGGGACCTT
*HvHMA3*	*HORVU5Hr1G094430.8*	ACATCGCCGTGAGGACAAGT	GCGTCTTGGACTTGCTCTGC
*HvHMA4*	*HORVU7Hr1G108890.1*	TCAGCCTAAGTCACAGAAGACATTG	CCTGGACGATTTCATCCTTGC
*HvHMA6*	*HORVU7Hr1G100160.2*	GCTAAGGCATCTATCGGTTCC	ATGCAGAACACTTTACTGCCTCT
*HvActin*	*HORVU1Hr1G002840*	TGGATCGGAGGGTCCATCCT	GCACTTCCTGTGGACGATCGCTG

**Table 2 plants-10-01849-t002:** Physicochemical properties and subcellular localization of HMA proteins in barley.

Gene Name	Protein Number	ORF (aa)	MW (Da)	PI	Subcellular Localization	Hydrophilicity Index
*HvHMA1*	A0A287XH51	1009	108464.81	6.68	Plasma membrane	−0.090
*HvHMA2*	A0A287TV87	987	106112.96	5.58	Plasma membrane	0.171
*HvHMA3*	A0A287SBM8	765	80511.64	5.80	Plasma membrane	0.422
*HvHMA4*	A0A287XS00	1023	102819.64	5.27	Plasma membrane	0.268
*HvHMA5*	A0A287J245	672	73112.29	6.19	Plasma membrane	0.277
*HvHMA6*	A0A287XKH5	871	92584.78	7.59	Plasma membrane	0.106
*HvHMA7*	M0X9Y2	761	80246.05	5.36	Plasma membrane	0.223
*HvHMA8*	A0A287NAD1	1050	113188.47	5.00	Plasma membrane	0.193
*HvHMA9*	A0A287N6A1	1000	109664.92	5.72	Endoplasmic reticulum	0.171
*HvHMA10*	M0WLW4	946	100370.69	7.04	Plasma membrane	−0.016
*HvHMA11*	A0A287FY78	981	107543.64	6.42	Plasma membrane	0.083
*HvHMA12*	A0A287E4C3	1039	114071.38	6.95	Plasma membrane	−0.042
*HvHMA13*	A0A287QI47	1033	111820.68	6.07	Plasma membrane	0.183
*HvHMA14*	A0A287EJS4	1049	113618.24	5.49	Endoplasmic reticulum	0.140
*HvHMA15*	M0WME5	962	105942.79	6.15	Plasma membrane	0.094
*HvHMA16*	A0A287DYP4	983	107457.12	6.10	Plasma membrane	0.063
*HvHMA17*	A0A287JG07	1010	106141.11	7.82	Plasma membrane	0.074
*HvHMA18*	A0A287RKR1	690	73747.50	6.73	Plasma membrane	0.219
*HvHMA19*	M0WL52	1020	110187.74	6.21	Endoplasmic reticulum	0.220
*HvHMA20*	A0A287Q5Y4	994	109774.70	7.45	Plasma membrane	0.142
*HvHMA21*	A0A287P785	1083	118116.93	5.49	Plasma membrane	0.034

## Data Availability

Not applicable.

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
