# Peer review of "Genome-Wide Identification of the HMA Gene Family and Expression Analysis under Cd Stress in Barley"

_plants, 2021, doi:10.3390/plants10091849_

Round 1

Reviewer 1 Report

l 143 - more details on data curation would helpful
from line 144: While I think these info are potentially interesting, I think the authors should mention why they think they are particular relevant for this protein group

3.2. How robust is your tree - did you try alternative tree building algorithm - NJ is usually only a crude method.

Expression and phylogeny - did you find that more closely related protein members are co-expressed?

prtoeins 10,1,6 and4 as well as 7,16,12 and closely neighbouring on the chromosomes. Did you check whether they emerged from duplications. 

I would suggest to conduct dN/dS analysis with these protein members to identify potential novel residues that could hint at novel functions.

Reviewer 2 Report

This manuscript presents the study of the heavy metal ATPase (HMA) gene family in barley using bioinformatics tools as well as the investigation of the expression level of some genes from this family under Cd stress in barley on the early stage of ontogenesis.

The topic of manuscript fits within the scope of the Plants journal and it is an interesting contribution to scientific knowledge in the field of plant physiology and plant molecular biology, in particular, of studies of plant resistance to heavy metal stress. In addition, the results of the presented research are of interest for producing new barley varieties with high resistance to Cd stress or for regulating the content of this pollutant in barley plants.

The manuscript is well prepared, nicely organized and written. However, the paper needs some revisions.

The major comment is related to the Discussion of the results. In the present version of the manuscript, most of the discussion is a repetition of the description of the results from the previous section. I would like to invite authors to discuss more in detail the role of different HvHMA genes in transport, translocation and detoxification of Cd in barley plants, as well as the possible impact of plant age and the level of Cd-stress (time of exposure and the concentration of Cd in medium) on the expression of different HvHMA genes. I suggest that the authors read the following articles to enhance their discussion of results:

1) Mills RF, Peaston KA, Runions J, Williams LE (2012) HvHMA2, a P1B-ATPase from Barley, Is Highly Conserved among Cereals and Functions in Zn and Cd Transport. PLoS ONE 7(8): e42640. https://doi.org/10.1371/journal.pone.0042640

2) Kaznina, N. M., Titov, A. F., Topchieva, L. V., Batova, Y. V., & Laidinen, G. F. (2014). The content of HvHMA2 and HvHMA3 transcripts in barley plants treated with cadmium. Russian journal of plant physiology, 61(3), 355-359. https://doi.org/10.1134/S1021443714030066

3) Wang, X.-K.; Gong, X.; Cao, F.; Wang, Y.; Zhang, G.; Wu, F. HvPAA1 Encodes a P-Type ATPase, a Novel Gene for Cadmium Accumulation and Tolerance in Barley (Hordeum vulgare L.). Int. J. Mol. Sci. 2019, 20, 1732. https://doi.org/10.3390/ijms20071732

4) Mikkelsen MD, Pedas P, Schiller M, Vincze E, Mills RF, Borg S, et al. (2012) Barley HvHMA1 Is a Heavy Metal Pump Involved in Mobilizing Organellar Zn and Cu and Plays a Role in Metal Loading into Grains. PLoS ONE 7(11): e49027. https://doi.org/10.1371/journal.pone.0049027

5) Kintlová, M., Vrána, J., Hobza, R., Blavet, N., & Hudzieczek, V. (2021). Transcriptome Response to Cadmium Exposure in Barley (Hordeum vulgare L.). Frontiers in plant science, 12. https://doi.org/10.3389/fpls.2021.629089

Please add also some relevant references to the phrase “… although there are few reports on the barley HMA gene family” in Introduction.

Minor comments:

1) Abstract:

1.1. The abbreviation LEA (L17) is not clear.

1.2. I suggest to add the names of genes in the sentence “Among the five HvHMA genes, three (???) were upregulated and two (???) were downregulated following Cd treatments”.

2) In Introduction: Please add some information about the tolerance or sensibility of barley plants to Cd stress. And about the toxic effects on barley plants caused by cadmium stress.

3) In Subsection 3.1: Please indicate how many plants were used for experiments in each group.

4) Subsection 2.3.3. Expression profiling of the barley HMA family and candidate gene selection, L119-125: Please add the name and reference to scale of growth stages of barley used in research. Why did not the authors use one of the more common scales such as BBCH-scale, Feekes scale or Zadoks scale?

5) Figure 6. What test was used for detecting the significance (t-test, Tukey test or some other)? Was the significance of the differences between variants 50 µmol/L and 100 µmol/L assessed? This could be helpful when discussing the results.

Round 2

Reviewer 1 Report

thank you for your responses

l 64 change "was lack" to "is lacking"

l65 family members were identified in the barley genome

l85 replace "quickly" with "immediately"

l86 in triplicates

Reviewer 2 Report

The authors improved the manuscript well. I am satisfied with all of the authors' responses to comments.